# Antibodies as Models and Tools to Decipher *Candida albicans* Pathogenic Development: Review about a Unique Monoclonal Antibody Reacting with Immunomodulatory Adhesins

**DOI:** 10.3390/jof9060636

**Published:** 2023-05-31

**Authors:** Jordan Leroy, Karine Lecointe, Pauline Coulon, Boualem Sendid, Raymond Robert, Daniel Poulain

**Affiliations:** 1CNRS, UMR 8576, UGSF—Unité de Glycobiologie Structurale et Fonctionnelle, University of Lille, F-59000 Lille, France; jordan.leroy@chu-lille.fr (J.L.); karine.lecointe@univ-lille.fr (K.L.); 2INSERM U1285, University of Lille, F-59000 Lille, France; 3CHU Lille, Laboratoire de Parasitologie-Mycologie, F-59000 Lille, France; pauline.coulon@chu-lille.fr; 4Kalidiv ZA, La Garde Bâtiment 1 B, Allée du 9 Novembre 1989, F-49240 Avrillé, France; raymond.robert@kalidiv.com

**Keywords:** monoclonal antibody, immunity, *Candida albicans*, diagnosis

## Abstract

Candidiasis, caused mainly by *Candida albicans*, a natural commensal of the human digestive tract and vagina, is the most common opportunistic fungal infection at the mucosal and systemic levels. Its high morbi–mortality rates have led to considerable research to identify the molecular mechanisms associated with the switch to pathogenic development and to diagnose this process as accurately as possible. Since the 1980s, the advent of monoclonal antibody (mAb) technology has led to significant progress in both interrelated fields. This linear review, intended to be didactic, was prompted by considering how, over several decades, a single mAb designated 5B2 contributed to the elucidation of the molecular mechanisms of pathogenesis based on β-1,2-linked oligomannoside expression in *Candida* species. These contributions starting from the structural identification of the minimal epitope as a di-mannoside from the β-1,2 series consisted then in the demonstration that it was shared by a large number of cell wall proteins differently anchored in the cell wall and the discovery of a cell wall glycoplipid shed by the yeast in contact of host cells, the phospholipomannan. Cytological analysis revealed an overall highly complex epitope expression at the cell surface concerning all growth phases and a patchy distribution resulting from the merging of cytoplasmic vesicles to plasmalema and further secretion through cell wall channels. On the host side, the mAb 5B2 led to identification of Galectin-3 as the human receptor dedicated to β-mannosides and signal transduction pathways leading to cytokine secretion directing host immune responses. Clinical applications concerned in vivo imaging of *Candida* infectious foci, direct examination of clinical samples and detection of circulating serum antigens that complement the Platelia Ag test for an increased sensitivity of diagnosis. Finally, the most interesting character of mAb 5B2 is probably its ability to reveal *C. albicans* pathogenic behaviour in reacting specifically with vaginal secretions from women infected versus colonized by this species as well as to display higher reactivity with strains isolated in pathogenic circumstances or even linked to an unfavourable prognosis for systemic candidiasis. Together with a detailed referenced description of these studies, the review provides a complementary reading frame by listing the wide range of technologies involving mAb 5B2 over time, evidencing a practical robustness and versatility unique so far in the Candida field. Finally, the basic and clinical perspectives opened up by these studies are briefly discussed with regard to prospects for future applications of mAb 5B2 in current research challenges.

## 1. Introduction

Candidiasis (either mucocutaneous or systemic) is an opportunistic infection caused mainly by *Candida albicans*, a yeast and natural commensal of the human digestive tract and vagina [1,2,3]. Unlike true pathogens, the presence of yeasts alone does not indicate their pathogenic character, which depends essentially on the susceptibility of the host and the expression of pathogenicity factors by some yeast strains [4]. It is therefore essential to have markers associated with pathogenicity to understand the mechanisms of infection [5,6] and to diagnose these infections as accurately as possible [7]. The development of hybridoma technology/advent of monoclonal antibodies (mAbs) led to considerable progress in answering some of these questions [8]. Among the mAbs developed, one generated in the mid-1980s (mAb 5B2) [9] allowed us to make considerable progress in understanding the mechanisms of pathogenesis and also contributed to the diagnosis of candidiasis and the tracing of more pathogenic strains. This unique contribution is based on more than 50 basic and clinical publications.

This review aims to discuss the linear nature of the development and use of this mAb in order to better understand the pathogenesis of candidiasis and its diagnosis. Information from current research suggests that many discoveries still have to be made in this field.

## 2. Basic Studies

### 2.1. Immunochemical Identification of mAb 5B2 Epitopes, Their Carrier Molecules and Genes Involved in Their Synthesis

Unlike many mAbs generated from the immunisation of mice with killed yeasts or molecules extracted from yeasts, mAb 5B2 was generated from an experimental infection in a rat [9].

As the epitope was found to be present in mannan, a major *C. albicans* cell wall polysaccharide both quantitatively and qualitatively, its precise nature was determined through a series of immunobiochemical studies concerning this molecule. These studies consisted of sequential depolymerisation and nuclear magnetic resonance analysis and were based around the impressive pioneering structural studies conducted in Japan by Prof. Suzuki’s group on *Candida* mannans or, more exactly, phosphopeptidomannans (PPMs), focusing on the so-called antigenic factors allowing Candida species or serotype identification by direct agglutination [10,11,12]. The 5B2 epitope was identified among the β-1,2-linked mannoside series [13,14,15,16,17]. These structures are rare in the living world and are expressed in large quantities in the most pathogenic species of the genus *Candida*, namely *C. glabrata* and *C. tropicalis*, which are isolated second and third, respectively, after *C. albicans* from human disease. A considerable amount of information on β-1,2 oligomannosides has been gathered by S. Suzuki’s group according to their relation with antigenic factors [18] and expression in different strains depending on growth conditions showing important changes related to pH [19] and temperature [20], triggering the yeast to hyphal transition [21]. An impressive synthesis gathering all this information was presented in a review published in 2012 [12]. The precise identification of the 5B2 epitope showed reactivity against a minimal degree of polymerisation of two β-1,2 linked mannoside residues, thus corresponding to antigenic factor 5 and in part antigenic factor 6 [22].

Western blot profiles revealed that the 5B2 epitope was expressed on a large number of *C. albicans* molecules [23] distributed among very high relative molecular weight polydispersed material containing large amounts of polysaccharides and bands with lower molecular weight and better resolution in gels, corresponding to mannosylated proteins. These patterns were observed for several other pathogenic *Candida* species, with each having specific mapping of the 5B2 epitope [24]. In turn, an analysis of these profiles led to the discovery of concanavalin A-unreactive glycoconjugates with no protein moiety and to the identification of a glycolipid we named phospholipomannan (PLM) [25]. PLM was then studied extensively and characterised as a cell wall molecule belonging to the manno-inositol-phosphoceramide family [17,26,27,28,29].

Refinement of the methods of extraction of *C. albicans* cell wall mannoproteins based on the knowledge of their mode of insertion (PPM, glycosylphosphatidylinositol (GPI)-anchored proteins with internal repeats (PIR proteins and secreted proteins)) led to a complete mapping of the molecules containing the 5B2 epitope, among which were well known “proteins” described as virulence factors, including Hwp1 and HSP 70 [30]. mAb 5B2 mapping of β-mannose epitopes in the mannoproteins of different cell wall fractions of mutants defective in the map kinase pathway revealed its importance in regulating the exposure of different surface anomery and modulation of the immune response [31].

Finally, mAb 5B2 contributed uniquely to the discovery and identification of the nine members of the gene family encoding β-mannosyl transferases (BMTs 1–9), responsible for the sequential addition of β-Mans on different carrier molecules. This has significantly improved our understanding of the β-Man biosynthetic pathway in *C. albicans* [32]. Further definition of BMT functions was achieved through sequential deletion in the strain BWP17 by PCR gene targeting. Briefly, β-Man transfer is under the control of BMTs 1 and 3 for the acid-stable fraction of PPM and BMTs 2, 3 and 4 for the acid-labile fraction. None of these four enzymes act on PLM, nevertheless BMTs 5 and 6 are specifically involved in the β-mannosylation of PLM. Concerning the O-mannosylation of cell wall mannoproteins, this depends on BMTs 1 and 3 [33,34,35,36]. Gene deletion of BMTs 7–9 was inconclusive and has not been the subject of specific studies to date. Only transcriptome analysis carried out to decipher *C. albicans* iron homeostasis mentions BMT 7 and BMT 9 in alternative genetic programs adapting to blood stream versus gut environments [37,38].

Recent developments in this area of research concern a highly pathogenic *Candida* species that emerged simultaneously in different countries worldwide with the characteristics of antifungal resistance and high virulence, leading to high rates of mortality among patients in intensive care units. This species, *C. auris*, was also shown to synthesise β-Mans [39].

Analysis of the distribution of the mAb 5B2 epitope confirmed high expression in some strains, placing *C. auris* in the same group as *C. albicans*, *C. tropicalis* and *C. glabrata*, namely the most pathogenic *Candida* species (Leroy et al., unpublished data to date). Experimentally, it has been shown that an IgG3 mAb specific for a β-1,2 linked mannotriose described as protective against *C. albicans* [40] also protected mice against *C. auris* infection [41]. Addressing the general topic of the interplay between *Candida* and human antibodies, an impressive study demonstrated that *C. albicans* was able to shape the antibody repertoire though CARD9-dependent induction of host-protective antifungal IgG, including against *C. auris* [42]. On the fungal side, a mechanism of yeast surface modulation of antigens, including β-mannosides, able to direct macrophage responses was discovered to be under the control of *C. albicans* mitochondrial proteins [43]. In our view, these fascinating findings from this area of research deserve to be highlighted.

### 2.2. Cytological and Histopathological Analysis of mAb 5B2 Epitope Expression

In parallel to the identification of these molecules, mAb 5B2 was involved in studies aimed at localising the expression of 5B2 epitopes. At the population level, direct immunoperoxidase staining of *C. albicans* colonies grown on agar revealed an unforeseen complex expression of the epitope distributed according to different sectors [44]. Immunofluorescence and confocal microscopy studies revealed a complex expression due to the multiplicity of molecules differently expressing the epitope on yeasts and hyphal forms at a given time in the cell cycle [45]. Among these, shedding of PLM from the cell wall on contact with host cells was demonstrated unambiguously [46,47]. mAb 5B2 was used early on in histopathological studies to assess the presence of invasive foci of *Candida* in different clinical situations and experimental models [48]. At the gut level, it was used to assess colonisation by yeast species expressing β-mannosides and to identify host and yeast backgrounds modulating intestinal interactions [49,50].

At the ultrastructural level, transmission electron microscopy studies on *C. albicans* ultrathin liquid nitrogen frozen sections probed with 5B2 directly coupled to gold particles showed its localisation within cytoplasmic vesicles merging with the cell membrane and crossing the cell wall through channels [51,52]. The merging of these channels at the cell wall surface corresponded to the “patchy” distribution of β-Man epitopes reported by other authors in immunofluorescence studies [53].

### 2.3. Contribution of mAb 5B2 to the Analysis of the Immunological Interface between C. albicans and 5B2 Epitopes or Molecules Expressing These Epitopes—Identification of the Ligands and Triggering Consequences

As a counterpart to the identification of the 5B2 epitope and the *C. albicans* molecules expressing these specific motifs, mAb 5B2 was involved in experiments to assess the properties of these molecules. In these immunobiological studies, mAb 5B2 was the probe used as a reference. One of the major findings was the identification of Galectin-3 (Gal-3) as the mammalian receptor for β-mannosides [45,46,47,54,55,56,57,58,59,60,61,62,63]. Although research on Gal-3, known as Mac2 antigen, was limited at this time [60], Gal-3 is now established as a lectin playing an important role in numerous human diseases; its important pleiotropic roles have generated more than 5000 papers. Interestingly, regarding *Candida* pathogenesis, circulating levels of Gal-3 appear to be markers of both infection and inflammation [64], a duality compatible with its ability to predict the recurrence of Crohn’s disease after surgery associated with persistent inflammation and *C. albicans* colonisation [65]. Regarding relative importance of Gal-3 among other *C. albicans* receptors, its discovery at the end of the 1990s was among the first of a long list of what has been called later Pathogens Recognition Receptors (PRRs) as a counterpart of fungal Pathogen Associated Microbial Patterns (PAMPs). Only mannose receptor was identified at this time [66,67], as early as in the 1980s, so that differentiating at this time a and β mannose binding was important. To our knowledge, apart from MBL, which was described simultaneously in 2000 [68], other major receptors were identified later: Dectin-1 in 2001 [69,70,71]; first TLRs in 2002 [72,73]; DC-Sign in 2003 [74]; Dectin-2 in 2006 [75,76]. Comprehensive reviews regularly synthesized the importance of these interactions in shaping the host immune response [77,78,79]. Regarding the modulation liable to be linked to Gal-3, a reason to delve into this notion is that the presence of β-1,2-mannosides in N-linked mannan reduces the production of inflammatory cytokines by dendritic cells [80]. This seems of pathophysiological relevance since the morphological transformation of Candida conidia into hyphae form is characterized by a decrease in the amount of phosphodiesterified acid-labile β-1,2-linked manno-oligosaccharides, whereas the amount of acid-stable β-1,2 linkage-containing side chains does not change [21]. The corollary to these findings may be that Candida variants reacting with mAb 5B2 are unable to induce a robust proinflammatory and an appropriate mechanism of antigen presentation. This should be noticed because dendritic cells show a unique pattern of C-lectin type receptors to carry out non-opsonic phagocytosis and produce cytokines that tailor the microenvironment in the immune synapsis to initiate the adaptive immune response. This disappearance of β-Man is of importance regarding unmaking of α-Mans. This topic relates to the so-called ASCA, human antibodies revealed by *S. cerevisiae* mannan allowing the diagnosis of Crohn’s disease but generated by *C. albicans* pathogenic phase [81]. These antibodies are indeed anti-oligomannose antibodies reacting with α-1,3 mannose at the non-reducing end of 2 or 3 α-1,2 linked mannose. Besides Crohn’s disease, ASCA are associated with a strong inflammatory response in a wide range of human diseases, most of which are associated with fungal dysbiosis and a *C. albicans* overgrowth [82].

### 2.4. Construction of mAb 5B2 Epitope Synthetic Analogues (β-1,2 Oligomannoside Series) and Analysis of Their Biological and Immunological Properties

With regard to the difficulties in producing β-1,2 oligomannosides by sequential degradation of *C. albicans* PPM, synthetic analogues were produced by chemical synthesis. mAb 5B2 was used to assess their conformity with natural products through the development of chemical synthesis steps. These synthetic probes mimicking adhesins were shown to prevent *C. albicans* colonisation in experimental mouse models [62]. When coupled to biotin sulfone, it was possible to detect specific antibodies either on microspheres by multi-analyte profiling technology (Luminex) or surface plasmonic analysis [83]. A recent paper involving mAb 5B2 used the same technology to determine the structural basis for protective epitope specificity and to discriminate the humoral responses of infected versus colonised patients [84].

In a similar approach aimed at identifying the structures mimicking β-Mans, a phage library expressing random peptides was screened with mAb 5B2. The application of this phage display methodology led to the isolation of peptides presenting with the same specificity as β-Mans in terms of adhesion and immunogenicity [85].

## 3. Clinical Studies

### 3.1. Diagnosis of Systemic Infections by the Detection of mAb 5B2 Epitopes

Alongside the mass of information accumulated at the basic/cognitive level, medical applications were investigated early on based on the assumption that the epitopes recognised were important for pathogenesis and that they were consistently expressed.

The first step consisted of demonstrating the presence of 5B2 epitopes circulating in the serum of experimentally infected animals and patients in order to prove the growth of yeasts in tissues. The technology used was immunogold silver staining (IGSS), an elegant, sensitive and highly specific method developed from the direct coupling of mAbs to gold particles used for ultrastructural localisation of antigens [48,86].

Another investigation concerned the ability of mAb 5B2 to reveal *Candida* infectious foci in living organism by radioimaging. In an innovative study at the time, the coupling of mAb 5B2 to iodine 131 demonstrated that it revealed pathognomic features of organ involvement in infected animals [87]. The ability of mAb 5B2 to detect antigens released by *Candida* in the sera of infected patients that were not detected by mAb EB-CA1 used in the *Candida* Platelia antigen test increased the sensitivity of antigen detection for rapid diagnostic purposes [88,89]. An immunomagnetic separation method involving magnetic beads coated with 5B2 was shown to be effective at concentrating *Candida* from blood [90,91]. From a technological point of view, the multiplicity of methods involving mAb 5B2 for immunodetection (IGSS, radiolabelling, coupling to latex particles, sandwich ELISA) has demonstrated the robustness and reliability of this probe.

### 3.2. Analysis of mAb 5B2 Epitope Expression as an Epidemiological Marker of the Virulence of C. albicans Strains

Flow cytometry analysis of strains grown in different conditions demonstrated the extent of regulation of surface β-mannoside expression modifying virulence properties [92], in concordance with early investigations with mAb 5B2 showing its ability to differentiate *C. albicans* strains according to the circumstances of their isolation (i.e., commensal vs. pathogenic situations). Indeed, when colonies isolated in the clinical mycology laboratory were tested by direct agglutination with purified mAb 5B2, strains isolated from blood cultures showed significantly higher expression of the epitope compared to strains isolated from other sites. This property was confirmed regardless of the geographical area of isolation (Lille, France; D. Poulain or Leicester, UK; F. C. Odds) [44]. Thus, mAb 5B2 appeared to be a possible marker of the pathogenic behaviour of *C. albicans*. Almost three decades later, this ability was confirmed in a larger collaborative study; 385 strains were studied by ELISA to assess the level of mAb 5B2 surface expression. Impressively, higher expression of the mAb epitope was significantly associated with higher mortality [93].

### 3.3. mAb 5B2 in the Particular Setting of Vaginal Infections

The vaginal mucosa is the clinical site where information about the pathogenic behaviour of *C. albicans* strains would be important to assess since *C. albicans* is both a frequent commensal and an opportunistic pathogen at this site. Vulvovaginal candidiasis occurs in 75% of women during their lifetime and up to 10% of women will suffer from recurrent infections that are particularly difficult to eradicate.

An initial study consisted of testing strains isolated from the vagina of symptomatic or asymptomatic women using latex particles according to a methodology developed by R. Robert [94]. This study revealed significantly higher agglutination scores in women exhibiting clinical signs of vaginal candidiasis (unpublished results; Catherine Bernard, Medical doctor thesis, 1995, Medical School, Lille, France).

These results prompted a second collaborative study assessing the reactivity of antigens released by *C. albicans* into vaginal fluid using an immunochromatographic test. According to the principles of the test, only molecules bearing 5B2 epitopes are retained from the vaginal sample. These migrate until the presence of the immune complex is captured in a band, revealing a positive test. This test had a negative predictive value of 98.6%, a positive predictive value of 96.6% and an efficiency of 98% in differentiating 130 symptomatic women with vaginitis from 75 asymptomatic controls, irrespective of the number of yeasts isolated in each category [95]. This confirmed the ability of mAb 5B2 to preferentially reveal *C. albicans* pathogenic behaviour.

In order to synthetically illustrate and complement this review, two addendums are provided. Figure 1 schematizes the different complementary achievements gained over time in basic and clinical research by using mAb 5B2. Table 1 lists the technologies involving mAb 5B2, emphasizing the unique robustness and flexibility of this probe.

## 4. Conclusions

Among the large number of mAbs generated against *C. albicans* to date, mAb 5B2 has been established as the tool that has made the most significant contribution to the complex analysis of *C. albicans* β-Man virulence factors. Concomitantly, it is also able to detect circulating antigens with diagnostic relevance in coherence with its epidemiological/pathophysiological ability to react more effectively with pathogenic species and pathogenic strains. To conclude on some perspectives concerning the use of mAb 5B2, it is clear on the clinical side that its ability to diagnose VVC, which is a worldwide medical and economic problem [98], should deserve to be confirmed by larger studies. Concerning more basic investigations, the balance between expression of b- and a-mannosides mAb 5B2 reveals probably one of the more interesting characters of *C. albicans* flexibility which has been overseen. The previously discussed balance between pro-inflammatory and anti-inflammatory responses triggered by this α/β shift perfectly fits with the elegant concept of the damage network proposed by A. Casadeval and A.L Piroski [99] to explain the different clinical features of candidiasis. In this respect, mAb 5B2 differences in *C. albicans* strains isolated from pathophysiological niches recall old epidemiological studies on serotypes A and B [10,11]. Large scale clinico-epidemiological mAb 5B2 typing of strains isolated in different pathological circumstances, i.e., IBDs versus invasive, would be worthwhile to assess the validity of this hypothesis.

## Figures and Tables

**Figure 1 jof-09-00636-f001:**
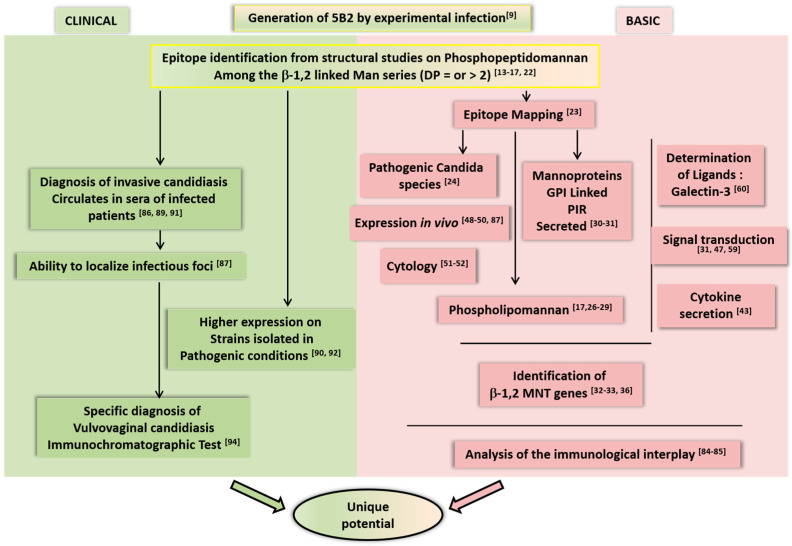
Schematized representation of the different achievements in basic and clinical research gathered over time by the use of mAb 5B2. The numbers in superscripts correspond to the studies described and referenced in the text.

**Table 1 jof-09-00636-t001:** List of technologies involving mAb 5B2 depending on the objectives of the different studies reported in the review.

Technologies	Objective	Achievement	References
BASIC IMMUNOCHEMISTRY/IN VITRO ANALYSIS			
**ELISA inhibition** *C. albicans* phosphopeptidomannan as substrate and inhibition by oligoammnosides released by sequential depolymerization	To identify mAb 5B2 epitope	Identification of mAb 5B2 epitope among the β-mannopyranosyl series	[13,14,15,16,17]
**Reactivity with neoglycolipids** Development of method for constructing neoglycolipids to study the roles of oligomannoside sequences in the antigenicity of *C. albicans* phosphopeptidomannan		Confirmation with monospecific rabbit polyclonal IgG	
**Flow cytometry**	To differentially probe *C. albicans* surface glycans over a yeast population To study the remodeling of the *C. glabrata* cell wall in the gastrointestinal tract	Ability of *C. albicans* to regulate its glycan surface expression and therefore modify its virulence properties Evidence that inflammation of the gut alters the microbial balance and remodels *C. glabrata* cell wall	[50,92]
**Phage display**	To seek for peptides sequences mimicking 5B2 epitope	Identification of a peptide sequence mimicking antigenicity and immunogenicity of the β-oligomannose epitope	[85]
**Plasmon magnetic resonance**	To finely differentiate mAb B6.1 and mAb 5B2 epitopes	In contrast to mAb B6.1, mAb 5B2 does not cross react with α-1,2 linked mannosides	[83,84]
**Indirect immunofluorescence** **confocal microscopy**	To analyze the complexitiy of β-1,2 Mannoside epitope expression at *C. albicans* cell wall surface	Heterogeneous and highly variable or patchy expression of 5B2 epitope at cell wall surface of different *C. albicans* growth forms	[1,46,52]
**Direct coupling to gold particles** **Electron microscopy on frozen *C. albicans* sections**	To analyze cell wall and cytoplasmic sub-cellular localization of mAb 5B2 epitope	Presence of β-Mans in vesicles merging with the cell membrane, moving from periplasmic space to cell wall fimbriaes following channels within the cell wall	[51,52]
**Western blotting**	To identify molecules expressing β-Man epitopes and members of the BMT gene family responsible for sequential addition of β-Man on these substates	Besides PPM, the epitope is shared by numerous mannoproteins	[23]
		High expression and specific patterns of distribution exist in *C. albicans*, *tropicalis glabrata*	[9]
		The epitope is shared by GPI anchored, PIR and secreted major *C. albicans* virulence factors	[30]
		The epitope allows characterization of a glycolipid named phopholipomannan, a member of the manno-inositol family with strong immunomodulatory properties	[29,32,45]
		Discovery of BMT gene family responsible for synthesis of β-mannosides composed of 9 members and precise definition of Bmts 1–6 activities depending on the acceptor molecule	[32,33,96]
**IN VIVO ANALYSIS**			
**Imaging after radio-iodination**	To detect in vivo Candida infectious foci by using of radioiodinated monoclonal antibody	The biodistribution of ^131^I radioactivity was greater in infected animals than in healthy animals and increased as a function of the number of CFU per g in each organ	[87]
**Immunogold silver staining histopathology**	To detect Candida species cells in tissue sections or during direct examination on glass slide	The method is sensitive with good definition, rapid and easy to perform. Long conservation of samples for collections	[48]
**CLINICAL APPLICATIONS**			
**Immunogold silver staining dot blot of animal and human sera**	To investigate the ability of mAb 5B2 to detect circulating antigens in sera of animals experimentally infected by *C. albicans* and in sera of patients with invasive candidiasis	The method, easy to perform, is adapted to the screening of large number of patients under serological surveillance	[86]
**Sandwich ELISA for immunocapture**	To improve the sensitivity of invasive candidiasis diagnosis by joint detection of α and β-mannosides	Provide evidence for different kinetics of β and α-Man circulation during experimental and human candidiasis	[97]
**Development of a method for immunomagnetic capture of Candida cells in blood**	To increase the sensitivity of blood culture	The method saves at least 24 h to obtain colonies by comparison to automated blood cultures	[91]
**Immunochromatographic test**	To diagnose vulvo-vaginal candidiasis	Point of care test adapted to the management of women with symptoms of vaginitis and differentiate colonization from infection	[95]
**EPIDEMIOLOGY/PATHOGENICITY MARKER**			
**Direct agglutination of isolated *C. albicans* strains**	To seek for differences in strain pathogenic potential	Higher expression in strains isolated in pathogenic vs. saprophytic conditions in human	[44]
**ELISA after coating of isolated *C. albicans* strains**	To investigate the correlation between yeast strain phenotypical features and patient outcome	Higher expression of β-Man detected by mAb 5B2 is related to increased patient mortality	[93]

## Data Availability

Not applicable.

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
