# Peer review of "Antibodies as Models and Tools to Decipher Candida albicans Pathogenic Development: Review about a Unique Monoclonal Antibody Reacting with Immunomodulatory Adhesins"

_jof, 2023, doi:10.3390/jof9060636_

Round 1
Reviewer 1 Report
This is a review detailing the historical time-course of the research devoted to
This is a review detailing the historical time-course of the research devoted to unveil the association of 1,2-mannan anomers with the pathogenic competence of Candida species.
Specific comments:
1. The authors highlight the relevant studies conducted by Prof. Suzuki, but I did not observe any appropriate quotation, when the paper by Shibata el al., Infect. Immun. 60:4100-4100, 1992, identified an epitope of antigenic factor 5 in mannans with β-1,2-linked oligomannosyl residues embedded in phosphodiesterified acid-labile chains and in N-linked acid-stable chains. I think this reference and a more detailed explanation of the structure of mannan chains are required.
2. A further reason to delve into this notion is that the presence of β -1,2-mannosides in N -linked mannan reduces the production of inflammatory cytokines by dendritic cells (Ueno et al., Medical Mycology 51:385-395, 2013). This seems of pathophysiological relevance since the morphological transformation of Candida conidia into hyphae form is characterized by a decrease in the amount of phosphodiesterified acid-labile β-1,2-linked manno-oligosaccharides, whereas the amount of acid-stable β-1,2 linkage-containing side chains does not change (Shibata et al., Biochem. J. 404 (3): 365–372, 2007). The corollary to these findings may be that Candida variants reacting with mAb 5B2 are unable to induce a robust proinflammatory and an appropriate mechanism of antigen presentation. This should be noticed because dendritic cells show a unique pattern of C-lectin type receptors to carry out non-opsonic phagocytosis and produce cytokines that tailor the microenvironment in the immune synapsis to initiate the adaptive immune response.
3. There seems to be an overrepresentation of contributions by the authors regarding the interaction of 5B2 epitopes with host cells and the role of galectin-3. Even this is properly documented, mannan chains have been found to be recognized at least by TLR receptors, dectin-2, and DC-SIGN. This should be considered since β -1,2-mannoside chains are surrounded by α-1,2-mannoside chains, not to mention the concomitant recognition of β-glucan chains by dectin-1.
4. Paying attention to these chemical and immunological aspects of mannans’ chemical biology may reinforce the significance of this review.
Reviewer 2 Report
This manuscript summarized the findings of previous basic and clinical studies on mAb 5B2 in Candida albicans infection, and could be considered a novel and valuable review.
My overall recommendation is acceptance, but there are still some modifications needed to improve the manuscript.
1. As the protagonist of the entire review, mAb 5B2 is not mentioned in the title and abstract, which can be very confusing for the reader.
2. Compared to the content of the entire manuscript, the abstract is very simple and unfocused, and there are many introductory sentences. It is recommended that the author rewrite the abstract to match the entire review.
3. The conclusion is too simple, without the authors' own insights and opinions, which makes the manuscript very "listy", so it is recommended to make some additions to the prospects for the future application of mAb 5B2 or its disadvantages.
Although there are no major grammatical and vocabulary errors, it is recommended to ask native English speakers to re-polish the manuscript to make it easier to understand.
